# FORMAL CONCEPTUAL VIEWS IN NEURAL NETWORKS

## ABSTRACT

Explaining neural network models is a challenging task that remains unsolved in its entirety to this day. This is especially true for high dimensional and complex data. With the present work, we introduce two notions for conceptual views of a neural network, specifically a many-valued and a symbolic view. Both provide novel analysis methods to enable a human AI analyst to grasp deeper insights into the knowledge that is captured by the neurons of a network. We test the conceptual expressivity of our novel views through different experiments on the ImageNet and Fruit-360 data sets. Furthermore, we show to which extent the views allow to quantify the conceptual similarity of different learning architectures. Finally, we demonstrate how conceptual views can be applied for abductive learning of human comprehensible rules from neurons. In summary, with our work, we contribute to the most relevant task of globally explaining neural networks models.

## 1 INTRODUCTION

Neural networks (NN) are known for their great performance in solving learning problems. However, these excellent results are almost always achieved at the price of human explainability. This problem is addressed in research and practice from different standpoints. There are calls to refrain from using NN for important problems and to rely on explainable methods, even if they give worse results in terms of accuracy (Rudin, 2019). The second major direction is to develop methods for explaining NN models. Such explanations can be classified as *local explanations*, i.e., why a particular data point was treated in a specific manner (Ribeiro et al., 2016), and *global* explanations, i.e., approaches for explaining the whole NN model. The latter can be achieved, e.g., by mapping the NN to an explainable surrogate. A common approach for locally explaining NN models is to highlight activation at some hidden layer (Fong & Vedaldi, 2018) or, if possible, project this inversely. For flat data, e.g., images, this is a viable approach since an essential explanatory component, the human, can be integrated into the process. This is not the case for high-dimensional or complex data. Global approaches are more difficult, in particular for high-dimension, and therefore less frequent. A typical idea is to find an (explainable) surrogate for a NN, e.g., symbolic regression (Alaa & van der Schaar, 2019).

We answer to the still growing interest for *global explanations procedures for NN models* by introducing a novel intermediate space, called (symbolic) conceptual views. We demonstrate how NN models can represented in these views and how surrogate training, e.g., with decision trees, can profit from this. We further demonstrate how to compare NN models, e.g., when derived from diverse architectures, using Gromov-Wasserstein (Mémoli, 2011) distance within the views. Moreover, we demonstrate how symbolic conceptual views can be used to represent NN models with formal concept lattices (Ganter & Wille, 1999) and profit from its human-centered approach for explainable data analysis. Finally, we show by an application of subgroup discovery how human-comprehensible propositional statements can be derived from NN models with the use of background knowledge. This allows us to extract global rules in form of propositional statements using the neurons of the NN.

## 2 RELATED WORK

Several approaches aim to provide insights or explanations into neural networks. Many of them highlight parts of the input that were relevant for a particular prediction (Ribeiro et al., 2016), so called *local explanations*. Those however, rely on the user's capability to comprehend input data representations. Hence, this approach is infeasible for higher dimensional learning problems. To overcome this limitation, the SOTA is to interpret models using *symbolic concepts*, an approach of

neuro→symbolic AI (Sarker et al., 2022). For example, Mao et al. (2019), Asai & Fukunaga (2018) and Fong & Vedaldi (2018) introduce methods which classify the inputs of a model to pre-defined concepts. Hence, they require manually created input representations for all pre-defined concepts, in contrast of extracting them automatically. Particularly successful is *TCAV* (Kim et al., 2018), which predicts the importance of user-defined concepts. The above are complemented by methods that automatically detect concepts for a given set of input/output pairs through identifying similar patterns of input samples at a given layer, e.g., *ACE* (Ghorbani et al., 2019). So far these methods do detect only particularly outstanding concepts. Recent works try to estimate to which extent a detected set of concepts is capable to approximate the model (Yeh et al., 2020). This approach, however, emphasizes classification performance and not explainability, i.e., concepts that are important for explanations may be omitted. This is in general true for surrogate based procedures that were not designed towards human comprehensibility (Alaa & van der Schaar, 2019). Moreover, a recent study shows that the translation of initial layers does often correlate with random layers or gradient detectors in the input (Adebayo et al., 2018). The most crucial downside of the automatic detection methods above is that although they provide symbolic concepts, these do not have to be interpretable. The overall principle of our approach is based on the fact that a substantial portion of the input data is aggregated and represented in the last hidden layer (Clark et al., 2019; Korbar et al., 2017).

A global interpretation of the NN needs a decoding into a human comprehensible symbolic view. A mathematical method for human comprehensible conceptualizations in the language of algebra is *formal concept analysis* (FCA) (Ganter & Wille, 1999; Wille, 1982). In particular its well-elaborated *conceptual scaling* theory (Ganter & Wille, 1989) provides an extensive tool-set to analyze NNs. This tool-set enables both the translation of neural representations into a symbolic space and further on the translation of this space into a human explainable space (Hanika & Hirth, 2022; 2021).

## 3 THE VIEWS OF NEURAL NETWORKS

We introduce in the following two notions of conceptual view of a neural network, in detail a *many-valued* and a *symbolic* view. Both provide novel methods to enable a human AI analyst to grasp deeper insights into the knowledge that is captured by the neurons. In addition to that the symbolic view facilitates the application of abductive learning procedures. This results in rules that allow to explain a NN by means of human comprehensible terminology, as well as, in terms of the neurons.

Let $N$ be the set of neurons of the last hidden layer of a NN. We interpret NNs as a function that maps input objects $g \in G$, that are represented as $g = (v_1, \ldots, v_m) \in \mathbb{R}^m$, to outputs in $[0, 1]^{|C|}$ for classes $C$. The parameter $m$ specifies the number of input features (see Figure 1). Naturally, we can interpret each neuron $n \in N$ as a function by itself from the input layer up to the activation of $n$, i.e., $n : \mathbb{R}^m \to \mathbb{R}$. The output neurons can be characterized analogously by a map $c : \mathbb{R}^{|N|} \to \mathbb{R}$. With $w_{i,j}$ we address the weights connecting the output neuron $c_i \in C$ with hidden neuron $n_j \in N$.

**Definition 1 (Many-Valued Conceptual View)** *Let NN be a neural network, $C$ its output classes and $N = \{n_1, \ldots, n_h\}$ the neurons of the last hidden layer. We define the* many-valued conceptual view *as $\mathcal{V} = (\mathbb{O}, \mathbb{W})$, where $\mathbb{O} \in \mathbb{R}^{|G| \times |N|}$ with value at $(i, j)$ equal to the activation $n_j(g_i)$, called* Object View, *and $\mathbb{W} \in \mathbb{R}^{|C| \times |N|}$ with value at $(i, j)$ equal to the weight $w_{i,j}$, called* Class View.

To give a short motivation: With the object view $\mathbb{O}$, we want to study the activation of the neurons $N$ given an object $g$. Complementary, with the class view $\mathbb{W}$, we investigate the relation of the neurons $N$ to the outputs $c \in C$ by their corresponding weights $w_{i,j}$. For example, we refer the reader to Figure 1, which depicts the object and class view (right) of the network (left). In this, we find that $n_k(o_t)$ is greater than $n_1(o_t)$, from which we infer that the relation of $o_t$ to $n_k$ is greater than $n_1$. We want to employ the just introduced views to comprehend the complete classification that is captured by a NN model. We can represent any object $g$ as a row in the object view matrix, i.e., $O(g) := (n_1(g), \ldots, n_h(g))$. Analogously, we can represent any class $c_i$ as a row in the class view matrix, i.e., $W(c_i) := (w_{i,1}, \ldots, w_{i,h})$. The outputs of the NN for class $c_i$ follow from the term $O(g) \cdot W(c_i) + b$, where $b$ is a bias. This can be rewritten as $|O(g)| \cdot |W(c_i)| \cos(O(g), W(c_i)) + b$ where $\cos(O(g), W(c_i))$ is the cosine value of the angle between $O(g)$ and $W(c_i)$. Thus, to understand the inner representation of the classes $C$ within the NN, it may be reasonable to grasp the objects and classes in the same space and classify objects using similarity measures. Using this approach we can introduce an *object-class distance map*

$d_\mathcal{V} : G \times C \to \mathbb{R}, (g, c) \mapsto d(O(g), W(c))$, where a sensible choice for $d$ is *cosine similarity* or the *Euclidean distance*. We will investigate both in Section 4.1. Hence, using $d_\mathcal{V}$ and similar distance maps for $G \times G$ and $C \times C$, on can derive a pseudo metric space $(G \cup C, \hat{d}_\mathcal{V})$. From this representation of $G$ and $C$ one can infer a simple classification map, e.g., by applying 1-NN classification.

**Similarity of neural networks**  Conceptual views enable a direct comparison of NNs. One can employ the Gromov-Wasserstein distance (Mémoli, 2011), as experimentally demonstrated in Section 4.2. We contrast our results with a baseline of model fidelity. We may note two important facts. First, our approach for similarity is comparable to the recent idea of relating neural networks to particular *kernel spaces* (Shankar et al., 2020; Lee et al., 2019). This enables us to study how objects are *hierarchically clustered* in such a space. We may stress that our notion does not consider how objects are mapped into this (kernel) space, but rather investigates the space itself. Second, the used GW distance is invariant with respect to permutations of the many-valued conceptual views.

### 3.1 SYMBOLIC VIEW

The next step is to scale the many-valued conceptual view into a symbolic space. We employ *conceptual scaling* (Ganter & Wille, 1989) from formal concept analysis (FCA), where data is represented in a *formal context* $\mathbb{K} = (G, M, I)$. There, $G$ is a set of objects, $M$ a set of attributes and $I \subseteq G \times M$ is an incidence relation, where $(g, m) \in I$ indicates that $g$ has attribute $m$. From $I$ arise two *derivation operators* $(\cdot)^I : \mathcal{P}(G) \to \mathcal{P}(M)$, with $A^I = \{m \in M \mid \forall g \in A : (g, m) \in I\}$, and analogously $(\cdot)^I : \mathcal{P}(M) \to \mathcal{P}(G)$, with $B^I = \{g \in G \mid \forall m \in B : (g, m) \in I\}$. Applying many-valued conceptual scaling to a many-valued data set yields a formal context. In this first attempt, we decided for *dichotomic scaling*, using thresholds for the object view $\delta_\mathbb{O}$ and the class view $\delta_\mathbb{W}$ (see Figure 1). As a final remark before we introduce the symbolic conceptual view on NN we want to point out a simple but powerful observation. The to be employed relational structure is invariant with respect to row- or column permutations in the related many-valued conceptual view (Definition 1).

**Definition 2 (Symbolic Conceptual View)** *Let* $\mathcal{V} = (\mathbb{O}, \mathbb{W})$ *the many-valued conceptual view of a NN and let* $\delta_\mathbb{O}, \delta_\mathbb{W}$ *be threshold values. We define the* symbolic conceptual view $\mathcal{V}_\mathbb{D} = (\mathbb{O}_\mathbb{D}, \mathbb{W}_\mathbb{D})$ *by*

$$\mathbb{O}_\mathbb{D} := (G, N \cup \bar{N}, I_\mathbb{O}), \text{ with } (g, n_j) \in I_\mathbb{O} :\Longleftrightarrow n_j(g) > \delta_\mathbb{O} \text{ and} \qquad \text{(Symbolic Object View)}$$
$$(g, \bar{n}_j) \in I_\mathbb{O} :\Longleftrightarrow n_j(g) \le \delta_\mathbb{O}$$
$$\mathbb{W}_\mathbb{D} := (C, N \cup \bar{N}, I_\mathbb{W}), \text{ with } (c_i, n_j) \in I_\mathbb{W} :\Longleftrightarrow w_{i,j} > \delta_\mathbb{W} \text{ and} \qquad \text{(Symbolic Class View)}$$
$$(c_i, \bar{n}_j) \in I_\mathbb{W} :\Longleftrightarrow w_{i,j} \le \delta_\mathbb{W}.$$

*We introduced with* $\bar{N} := \{\bar{n} \mid n \in N\}$ *a set of artificial symbols and used them as defined above.*

This definition allows for constructing human comprehensible explanations given a background ontology, e.g., in form of human annotations of the objects or classes. We exemplify that in Figure 1 using the formal context $\mathbb{S}_N$ that employs interpretable features $S_{m_1}, \ldots, S_{m_l}$. We provide more details in Section 5. Suitable threshold values $\delta_\mathbb{W}, \delta_\mathbb{O}$ depend on the architecture of the to be analyzed NN model. For example, if the activation function is ReLu, the neuron's co-domain is positive. Thus it becomes difficult to determine a reasonable $\delta$ for negative symbols $\bar{N}$, as studied in Section 4.3.

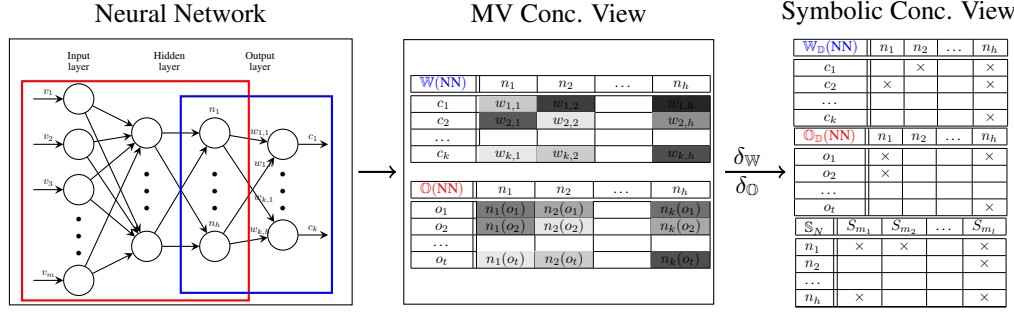

Figure 1: A simplified neural network drawing (left), its many-valued conceptual view (middle) and its symbolic conceptual view (right).

## 4 EXPERIMENTAL STUDY

We support our theoretical modeling of conceptual views, in particular the symbolic conceptual view, through an experimental study using common and well known data sets and NN models. First, we evaluate the suitability of the in Section 3 introduced pseudo metric space by a classification task. Second, we show how one may compare many-valued conceptual views (different NN models). Third, we demonstrate how to derive a human comprehensible representation for a NN model, that we can employ for explanations in Section 5. The code and trained models are available on GitHub.[1]

### 4.1 MANY-VALUED CONCEPTUAL VIEWS ON IMAGENET

We demonstrate that many-valued conceptual views are capable of capturing a large share of a NN model. For this, we use all twenty-four NN models from tensorflow that are trained on the ImageNet (Deng et al., 2009) data set. The object view is calculated using the test set, i.e., 100k images, of *ImageNet* used in the ILSVRC (Russakovsky et al., 2015) challenge. In Appendix, Table 6 we compiled basic statistics on these networks and our views. Although we report in columns two and four mean values and their standard deviation, we may stress that we do not consider the individual values to be normal distributed.

**Evaluate ImageNet Views** To evaluate the quality of our views, we compare a one-nearest-neighbor (1-NN) classifier on the in Section 3 introduced pseudo matric space $(G \cup N, \hat{d}_\mathcal{V})$ directly with the NN classification function on all 100,000 test images. In detail, we use model fidelity, i.e., we count the instances where the 1-NN outputs the same class label as the NN and normalize this number by the cardinality of the test set. The results are depicted in Table 1. We differentiate in our experiments between using cosine similarity and Euclidean distance within $\hat{d}_\mathcal{V}$.

**Observations** We find that the view model is capable of achieving high fidelity (see Table 1). The MobilNetV1 model is the only exception. Moreover, we can state that using the Euclidean distance is superior to the cosine similarity in all instances. This is in particular true for the ResNet models, where the difference is up to 0.6. Furthermore, for the EfficientNets we notice that there is an almost monotone relation between the number of neurons $N$ (last hidden layer) and the fidelity. All this together suggests that the many-valued conceptual view is meaningful and that classification functions that are based on the resulting psuedo metric space can be used as surrogates for the NN model.

### 4.2 SIMILARITY OF NEURAL NETWORKS

Based on many-valued conceptual view, we can derive for all NN models a pseudo metric space as introduced in Section 3. Hence, given the theory about metric spaces there are different approaches for comparing them. For example, one could compute the Gromov-Hausdorff (Mémoli, 2011) distance. However, due to the vast number of data points, any direct computation of the GH distance is infeasible. A different approach, which is still costly, but can be performed for a subset of the data, is the Gromov-Wasserstein (Mémoli, 2011) distance. In Figure 2 (right) we depict the individual distances for all considered models with respect to the class and object view. We employed ten percent of the test data set and applied a uniform probability measure on the data points, i.e., a

---

[1] https://github.com/FCA-Research/Formal-Conceptual-Views-in-Neural-Networks

Table 1: The fidelity between a NN model their MV conceptual view using 1-NN classifier. Find all twenty-four results in Appendix, Table 5

| Model | Euclidean | Cosine |
|---|---|---|
| VGG16 | 0.945 | 0.841 |
| DenseNet201 | 0.972 | 0.728 |
| MobilNetV1 | 0.575 | 0.449 |
| ResNet152V2 | 0.999 | 0.314 |
| EffB0 | 0.944 | 0.933 |
| EffB7 | 0.985 | 0.979 |

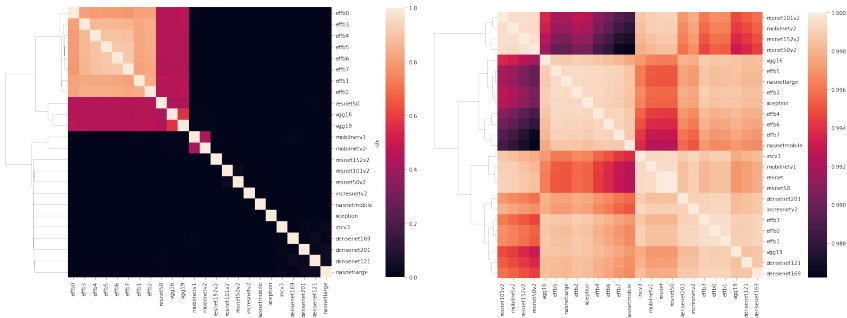

Figure 2: The similarity of twenty-four neural networks trained on the ImageNet data set. The base-line (left) is pair-wise fidelity between the employed models compared to a similarity using Gromov-Wasserstein distance on class view (right). Find the object view in Appendix, Figure 6.

normalized counting measure. We compare our analysis to a baseline that is derived from the fidelity measure (Figure 2, left).

**Observations**   From the pairwise fidelity diagram (Figure 2, left), we can infer that almost all models are very distinct with the exception of VGG16, VGG19, ResNet50 and the EfficientNet instances. In addition we find that the later models become more similar with increasing number of neurons. The similarity plot for the views are different from the fidelity plot. We can visually identify clusters of models. These clusters do often correspond with similar networks architectures. For example in the object view we observe two clusters. In the class view this clustering is finer.

## 4.3   SYMBOLIC CONCEPTUAL VIEW

In this section we study thoroughly the activation functions and the number of neurons for a reasonable determination of threshold values in order to compute meaningful symbolic conceptual views.

**Parameter Study**   To evaluate the influence of the choice of the activation function as well as the number of neurons, we trained one NN architecture several times on the *Fruits-360* (Mureşan & Oltean, 2017) data set. The used data set contains 67,692 images of 131 types of fruits or vegetables. The test set contains an additional 22,688 images. We train the architecture from the Fruits-360 experiment[2] using all procedure parameters from Mureşan & Oltean (2017) and modified the last two hidden layers. For the (last) hidden layer $N$ we vary the size $2^n$ between $2^4$ and $2^9$ with powers of two. For the layer before that we follow the common approach for smooth decrease in dimension, i.e., we chose $2^{\lfloor \frac{10+n}{2} \rfloor}$ with $4 \leq n < 10$ dependent on the last hidden layer. For activation functions, we studied the impact of *ReLu*, *Linear*, *Swish* and *Tanh* in all layers. For each parameter setting we trained ten models and computed their respective conceptual views for the test data set, see their distributions in Figure 3. Statistics on the quality of the computed views can be seen in Table 2. We tested these distributions against normalization of the column vectors in the views and can report that the reported results are invariant.

**Observations**   In general, we observe that the distributions for the object views differ quite largely among the different activation functions. From these we found that Tanh causes the most notable seperation of positive and negative values. We depicted the results for Tanh in  Figure 3 and for all other activation functions in Figure 7. Furthermore, we find that splitting with $\delta_{\mathbb{O}} = 0$ seems to be meaningful for all examples with respect to separation and symmetry. This split into two set of almost equal size. The same is true for $\delta_{\mathbb{W}}$. Apart from these values, we experimented in this and all following experiments with different approaches to determine thresholds, such as mean values, median values, median per neuron, as well as kernel-density estimation for bi-variate Gaussians. However, the split at 0 was favorable with respect to the achieved model fidelity. We report these scores for Tanh (Table 2) and for all activation functions in Appendix, Table 7. We conclude from our

---

[2]https://github.com/Horea94/Fruit-Images-Dataset

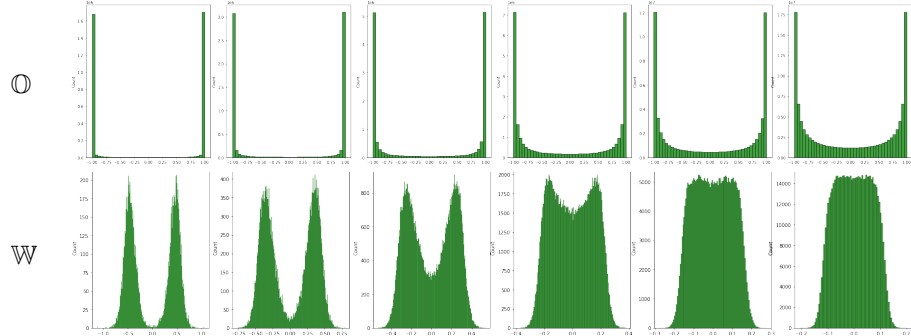

Figure 3: The value distributions for the object ($\mathbb{O}$) and class ($\mathbb{W}$) view for ten runs using the Fruits-360 data set and the **Tanh** activation function. The last hidden layer of size $2^4$ (first column) to $2^9$ (last column). The reader may find the plot for all activation functions in Appendix, Figure 7.

ablation study that the use of Tanh is suggested as well as $\delta_{\mathbb{O}}, \delta_{\mathbb{W}} = 0$. We acknowledge that the used data set might influence this choice (Bellido & Fiesler, 1993).

**Symbolic Conceptual Views of ImageNet**   Based on the just found parameters we computed the symbolic conceptual views for the ImageNet models. We report the results in Table 3. We found that the classes are uniquely represented (class separation equals 1), thus a perfection classification procedure is theoretically possible using the symbolic view. However, we observe that a direct application of 1-NN procedure using the binary vectors that arise from the symbolic view does result in very poor classification performance for ReLu. In contrast to that, the Swish based models achieved from mediocre to good results. Especially the larger (EfficientNet) NN models resulted in better symbolic views. A reason for the unfavorable results with ReLu might esteem from its positive co-domain, which hinders the construction of negated attributes $\bar{N}$ in our approach. Some selected distributions can be found in Appendix, Figure 8.

**Symbolic Conceptual Views on Fruit-360**   The twenty-four models in ImageNet employ ReLu and Swish activation functions only. Thus, we want to complement our experimental study on s.c. views with results for using Tanh, which we conduct again on the Fruits-360 data set. Hence, we trained five models, namely the base-line model from Mureşan & Oltean (2017), VGG16, ResNet50, IncV3, and EffB0, the latter initialized with the original ImageNet weights. With exception of the baseline, we added to each model three dense layers (including dropout layers with p=0.2) on top, that are sized 1024, 256, and 32. To all models we also added an additional layer of size 16. This reduction in size added in order to enable human explainability. The baseline model as well as all added layers employ the Tanh activation. The output (prediction) layer is a dense layer using *softmax* activation without bias. We used sparse categorical crossentropy as the loss function. All other relevant parameters for reproducing our results are drawn from the published baseline model.

The results in Table 4 show that all models have high accuracy on the test data set, while the four transfer learned models outperform the baseline. We want to stress that our predictive results are only

Table 2: Results for the ablation study on the influence of the activation function and number of neurons on the quality of the (Symbolic) object/class view. We measure the quality in terms of fidelity of 1-NN to the original model, see V-Fid and SV-Fid for the symbolic case. Find the full table in Appendix, Table 7.

| | $2^4$ | $2^5$ | $2^6$ | $2^7$ | $2^8$ | $2^9$ |
|---|---|---|---|---|---|---|
| **Tanh** | | | | | | |
| $\delta_{\mathbb{O}} = 0$ Split | 49.7/50.3 | 49.7/50.3 | 49.8/50.2 | 49.9/50.1 | 50.0/50/0 | 49.9/50.1 |
| $\delta_{\mathbb{W}} = 0$ Split | 49.9/50.1 | 49.8/50.2 | 49.8/50.2 | 50.0/50.0 | 49.9/50.1 | 50.0/50.0 |
| Model Acc | 90.5$\pm$ 0.8 | 94.3$\pm$ 0.5 | 94.7$\pm$ 0.5 | 94.9$\pm$ 0.4 | 95.0$\pm$ 0.4 | 94.8$\pm$ 0.3 |
| V-Fid | 98.3$\pm$ 0.5 | 99.5$\pm$ 0.1 | 99.7$\pm$ 0.0 | 99.7$\pm$ 0.0 | 99.8$\pm$ 0.0 | 99.8$\pm$ 0.0 |
| SV-Fid | 94.3$\pm$ 1.4 | 97.4$\pm$ 0.4 | 97.7$\pm$ 0.4 | 97.6$\pm$ 0.1 | 97.8$\pm$ 0.2 | 97.6$\pm$ 0.2 |

Table 3: The fidelity between six NN models and their symbolic conceptual view using 1-NN for classification. Find all values in Appendix, Table 8.

|  | Euclidean | Cos | Class Sep | Activation |
|---|---|---|---|---|
| VGG16 | 0.552 | 0.552 | 1.0 | ReLu |
| DenseNet201 | 0.000 | 0.000 | 1.0 | ReLu |
| MobilNetV1 | 0.036 | 0.036 | 1.0 | ReLu |
| ResNet152V2 | 0.000 | 0.000 | 1.0 | ReLu |
| EffB0 | 0.758 | 0.758 | 1.0 | Swish |
| EffB7 | 0.957 | 0.957 | 1.0 | Swish |

used to demonstrate that the model did fit to the classification problem. We find that both, Euclidean and cosine based 1-NN did perform well on the MV as well as the symbolic conceptual view spaces. In detail, we could not find significant difference between the representations. Moreover, we cannot identify significant differences in the classification performance with respect to the NN model. We also observed that an additionally trained decision tree classifier was unable to learn within the MV conceptual view representation. However, the same procedure applied to the symbolic view was capable of producing competitive classification results, a surrogate for the NN with very high fidelity.

**Symbolic Conceptual Views through FCA**   Each algebraic relation corresponds to exactly one natural order structuring of the data involved by means of the underlying Galois connections. Through FCA one can reveal this order by means of *formal concepts*, i.e., all $(A, B) \in \mathcal{P}(G) \times \mathcal{P}(M)$ such that $A^I = B$ and $B^I = A$. The sets $A$ and $B$ are called *extent* and *intent*, respectively. The reader may notice a strong resemblance to maximal bi-cliques. The set of all formal concepts of a context $\mathbb{K}$ is denoted by $\mathfrak{B}(\mathbb{K})$ and its elements are ordered by inclusion $\subseteq$ on the extent sets. The resulting order structure $\underline{\mathfrak{B}}(\mathbb{K}) := (\mathfrak{B}(\mathbb{K}), \subseteq)$ constitutes a complete lattice that can be visualized as a *line diagram*. Hence, applied to our analysis of NNs one may visualize the conceptual hierarchy learned by the network and its size serves as an upper bound for the number of learned concepts.

**Observations**   We computed the concept lattices for Base, ResNet, VGG16, IncV3, and EffB0 and find, that their sizes vary between 126487 (VGG16) and 134100 (IncV3) for $|N| = 16$, and between 3,498,829 (VGG16) and 3,803,799 (ResNet50). If we restrict our computation to $N$, i.e., omitting the artificially introduced negations $\bar{N}$, we find the concept lattice sizes decrease by one magnitude. In detail, between 5200 (VGG16) and 6573 (EffB0) for $|N| = 16$, and 150884 (EffB0) and 198152 (IncV3) for[3] $|N| = 32$. We compiled all values in Appendix, Table 9 Overall, we observe that all are similar in size and large and therefore cannot be visualized using a line diagram. We note that formal concepts are composed of combinations of features. The minimum number of encoded features is present in so called *meet-irreducible* (MI) elements, i.e., concepts where the extent of $(A, B)$ cannot be written as an intersection of extents from $\mathcal{B}(\mathbb{K}) \setminus \{(A, B)\}$. Hence, the number of MI elements is bound by the number of attributes in the context, which serves as a lower bound the concepts captured by the NN model.

Independent of the size does this translation to the realm of FCA enable the application of various knowledge-based methods, such as Description logic or Subgroup discovery, as investigated in Section 5. Within FCA we might consider to analyze cuts of the lattice, in particular those where we suspect problems in the representation. As we discovered in previous experiments, that *Apple Red*, *Pink Lady*, *Plum* and *Cherry* are indistinguishable by some symbolic representations (Table 4), one might want to "zoom" into those. We did this statistically with Figure 4 (top) and structurally with Figure 4 (bottom). In the former on can identify formal concept based similarities among the selected fruits and the number of their shared concepts. In the latter figure the reader can infer the hierarchical dependencies between the different fruits (objects) indicated in color.

## 5   ABDUCTIVE LEARNING OF PARTIAL EXPLANATIONS

Symbolic conceptual views enable the application of various logical methods to derive human-comprehensible (partial) explanations. We draw from this correspondence and construct a formal

---

[3]In this experiment we also studied the influence of omitting the layer of size 16.

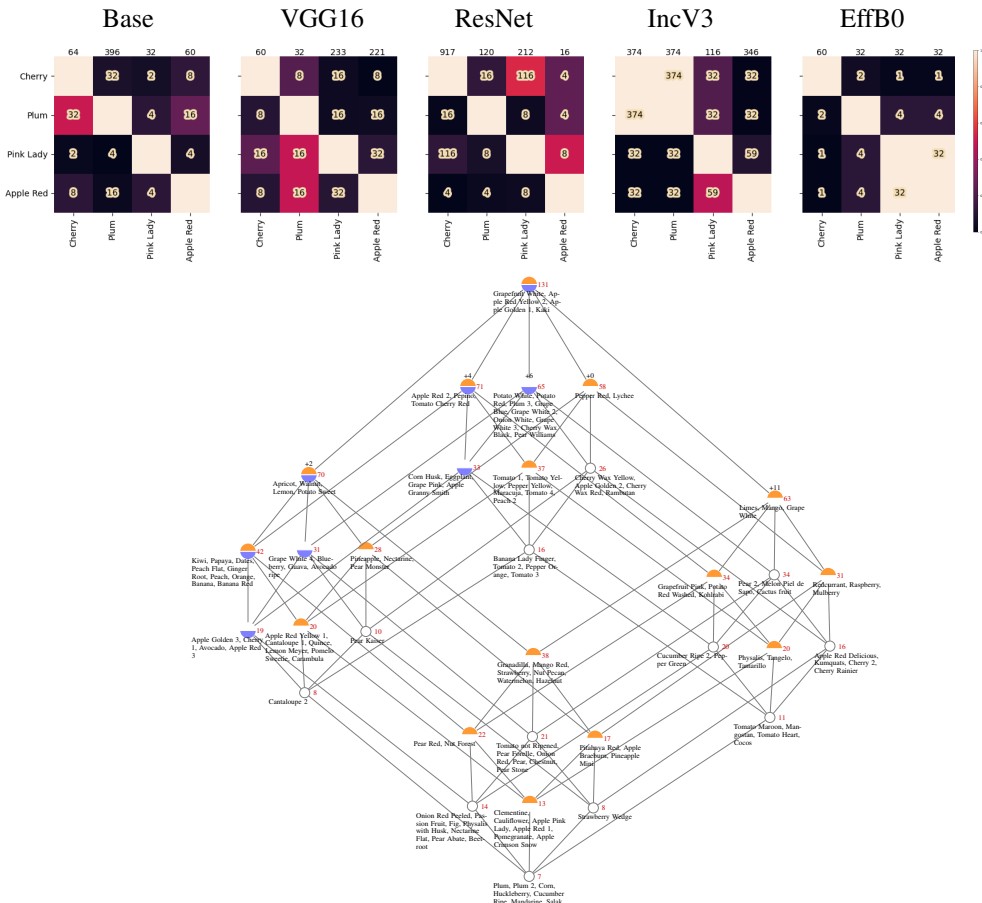

Figure 4: FCA results for *Apple Red*, *Pink Lady*, *Plum* and *Cherry* using views with positive attributes. **Top**: The columns labels of each heatmap displays the number of form. concepts. The number within a cell is the number of shared concepts between the related row/column fruits. Heat indicates its fraction. **Bottom**: Order-structured representation of the selected fruits. Formal concepts containing *Apple Pink Lady* and *Apple Red 1* are highlighted in Orange and *Cherry 1* is highlighted in blue.

context $\mathbb{C} = (C, S_M, I_{\mathbb{C}})$, where $S_M = \{S_{m_1}, \dots, S_{m_l}\}$ is a set of human-interpretable features that are known about the classes $C$, i.e., *background knowledge*.

**Definition 3 (Symbolic Interpretation)** *Given the s.c. view $\mathcal{V}_{\mathbb{D}} = (\mathbb{O}_{\mathbb{D}}, \mathbb{W}_{\mathbb{D}})$ of a NN, background knowledge $\mathbb{C}$, and a similarity relation $\sim$ on the classes $\mathcal{P}(C)$. Then is the formal context $\mathbb{S} = (N, S_M, R)$ with $(n, S_m) \in R :\Longleftrightarrow \{n\}^{I_{\mathbb{W}}} \sim \{S_m\}^{I_{\mathbb{C}}}$ the* symbolic interpretation *of the NN with respect to $\mathbb{C}$ and $\sim$.*

We require $\sim$ to be reflexive and symmetric but not necessarily transitive. The task for symbolic interpreting a NN is to deduce or infer $\sim$ using background knowledge, which is done in the next section. Given a symbolic interpretation for a NN we are able to express neurons in terms of the human-interpretable features $S_M$ by applying the incidence relation in $\mathbb{S}$, i.e., for all $n \in N$ one can compute $\{n\}^R$. Furthermore, if $\mathbb{S}$ is additionally equipped with propositional logic $\mathcal{L}(S_M, \{\vee, \wedge, \neg\})$ then FCA (Hanika & Hirth, 2022) also provides the means for expressing neurons in terms of propositional statements.

**Evaluate the Symbolic View Interpretation** We want to motivate how symbolic interpretations can be used to interpret neuron in terms of (human-comprehensible) features $S_M$ and vice versa. To demonstrate both cases, we analyze the symbolic view of the Fruit-360. The attributes $S_M$ we use visual features (1), such as shapes or colors, and we used the *Scientific classification* taxonomy (2)

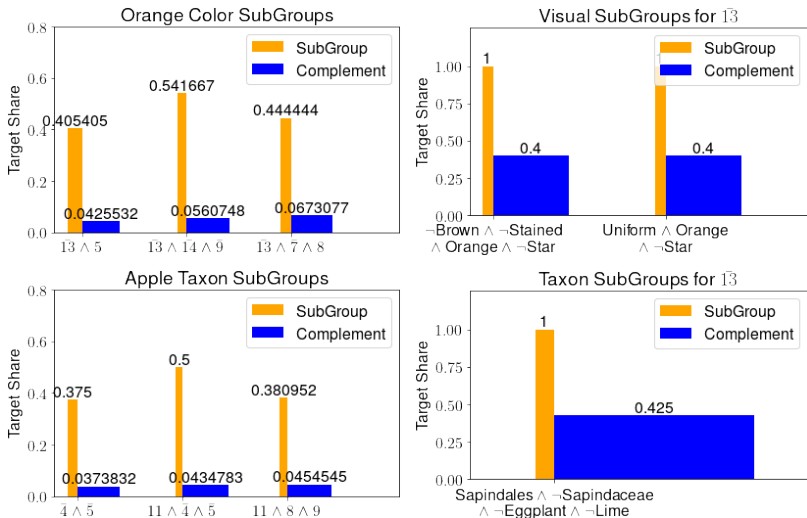

Figure 5: Exemplary results of the subgroup detection. Width indicates subgroup-size.

published in Wikipedia[4] for each fruit/vegetable. We combined the German and English Wikipedia articles in order to derive a data set as complete as possible. We infer the similarity relation in our experiment using *subgroup detection* (Herrera et al., 2010), as implemented in `pysubgroup`.

We depict four example results in Figure 5, where the taxons are given in the respective diagram titles, e.g., Apple, Orange. Diagrams on the left depict subgroups in terms of neurons, and, vice versa, on the right in terms of interpretable features. For both sides we find on the abscissa propositional statement combining the respective features.

From the high share (see ordinate) of the respective subgroups we can infer that the propositional statements using the neurons or $S_M$ features describe the taxons from adequately up to very good. In particular for the latter case (right) we see that the subgroups are pure, yet, not complete. To give two concrete statements: first, fruits that are not brown and not stained and not orange and not star shaped will use neuron $\bar{13}$ from $\bar{N}$. Second, if the neuron $\bar{13}$ and $\bar{14}$ and $\bar{9}$ are used by the NN this implies that the fruit is orange with confidence about 0.54. Using this method one can infer the similarity relation $\sim$ and provide an explanation framework.

## 6 LIMITATIONS AND CONCLUSION

The presented approach is novel and different former ideas with respect to three properties: first, we do not employ further hardly explainable methods, such as autoencoders. Second, our method is global by design. Third, conceptual views, as introduced in our work, do not require pre-defined concepts and their related input representations. We accomplished this by decoding both, the weights of all output neurons and the activations of the last hidden layer using symbolic conceptual views.

Our approach is limited by the necessary existence of multiple outputs. However, there are common approaches for splitting single outputs. Yet, a more significant limitation concerns the restriction to non-recursive architectures. Adapting our approach to such settings is probably possible, but requires a substantial adjustment to the definition of the views. Finally, our method requires for human-comprehensible explanations the existence of domain-specific background knowledge.

Apart from this we envision that the presented link of NN models to FCA using symbolic conceptual views allows for both the explainability of NNs as well as increasing the performance of NN surrogate learning procedures. Therefore, this beneficial research line should be further investigated and tested.

---

[4]See for example `https://en.wikipedia.org/wiki/Apple` in the right box.

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

## A  APPENDIX

In this appendix we provide more figures and table data for our experimental study.

Table 4: Five neural networks (using Tanh activation) and their (symbolic) conceptual views were captured by different surrogates (decision tree, 1-NN). We report their fidelity and accuracy. In symbolic conceptual view: IncV3 was unable to distinguish *Cherry 1* and *Plum*; EffB0 was unable to distinguish *Apple Red 1* and *Apple Pink Lady*.

| Model | Model | DTree | | Euclidean | | Cos | | |
|---|---|---|---|---|---|---|---|---|
| | ACC | ACC | Fid | ACC | Fid | ACC | Fid | |
| Baseline | 0.936 | 0.017 | 0.017 | 0.935 | 0.989 | 0.935 | 0.988 | |
| VGG16 | 0.988 | 0.017 | 0.018 | 0.988 | 0.998 | 0.988 | 0.997 | |
| ResNet50 | 0.989 | 0.018 | 0.018 | 0.989 | 0.998 | 0.989 | 0.997 | |
| IncV3 | 0.983 | 0.013 | 0.013 | 0.983 | 0.999 | 0.984 | 0.999 | |
| EffB0 | 0.984 | 0.007 | 0.007 | 0.984 | 0.998 | 0.983 | 0.984 | |
| Symbolic | | | | | | | | Class Sep |
| Baseline | 0.936 | 0.857 | 0.879 | 0.927 | 0.964 | 0.927 | 0.964 | 1.0 |
| VGG16 | 0.988 | 0.972 | 0.977 | 0.988 | 0.994 | 0.988 | 0.994 | 1.0 |
| ResNet50 | 0.989 | 0.952 | 0.957 | 0.988 | 0.996 | 0.988 | 0.996 | 1.0 |
| IncV3 | 0.983 | 0.975 | 0.988 | 0.984 | 0.997 | 0.984 | 0.997 | 0.992 |
| EffB0 | 0.984 | 0.938 | 0.934 | 0.984 | 0.996 | 0.984 | 0.996 | 0.992 |

Table 5: The fidelity between twentyfour neural networks and their object/class view using 1-NN for classification.

| Model | Euclidean | Cosine | Model | Euclidean | Cosine |
|---|---|---|---|---|---|
| VGG16 | 0.945 | 0.841 | ResNet101V2 | 0.995 | 0.466 |
| VGG19 | 0.942 | 0.842 | ResNet152V2 | 0.999 | 0.314 |
| IncV3 | 0.990 | 0.753 | IncResNetV2 | 0.999 | 0.983 |
| DenseNet121 | 0.978 | 0.737 | XCeption | 0.977 | 0.792 |
| DenseNet169 | 0.989 | 0.843 | EffB0 | 0.944 | 0.933 |
| DenseNet201 | 0.972 | 0.728 | EffB1 | 0.960 | 0.946 |
| MobilNetV1 | 0.575 | 0.449 | EffB2 | 0.969 | 0.957 |
| MobilNetV2 | 0.947 | 0.925 | EffB3 | 0.974 | 0.961 |
| NasNetMobile | 0.935 | 0.808 | EffB4 | 0.981 | 0.972 |
| NasNetLarge | 0.880 | 0.831 | EffB5 | 0.979 | 0.972 |
| ResNet50 | 0.954 | 0.800 | EffB6 | 0.982 | 0.976 |
| ResNet50v2 | 0.995 | 0.734 | EffB7 | 0.985 | 0.979 |

Table 6: The average weights $w_{i,j}$, object values $n_i(g)$ and the number of neurons $|N|$ and activation function $f$ of the last hidden layer of tensorflow imagenet models.

| Model | $\mathbb{W}$ - values | Bias | $\mathbb{O}$ - values | $|N|$ | $f$ |
|---|---|---|---|---|---|
| VGG16 | -5.359e-07 $\pm$ 0.008 | 1.404e-06 $\pm$ 0.191 | 0.679 $\pm$ 1.514 | 4096 | ReLu |
| VGG19 | -6.707e-07 $\pm$ 0.008 | -1.287e-05 $\pm$ 0.192 | 0.613 $\pm$ 1.402 | 4096 | ReLu |
| IncV3 | -3.808e-05 $\pm$ 0.034 | -0.0099 $\pm$ 0.308 | 6.025 $\pm$ 15.13 | 2048 | ReLu |
| DenseNet121 | 2.139e-08 $\pm$ 0.049 | -1.014e-07 $\pm$ 0.012 | 1.731 $\pm$ 4.603 | 1024 | ReLu |
| DenseNet169 | 1.456e-08 $\pm$ 0.039 | -1.038e-07 $\pm$ 0.012 | 1.675 $\pm$ 5.529 | 1664 | ReLu |
| DenseNet201 | 1.019e-08 $\pm$ 0.036 | -1.178e-07 $\pm$ 0.011 | 1.146 $\pm$ 4.167 | 1920 | ReLu |
| MobilNetV1 | -0.0001 $\pm$ 0.081 | -0.005 $\pm$ 0.744 | 0.435 $\pm$ 0.838 | 1024 | ReLu |
| MobilNetV2 | -3.138e-05 $\pm$ 0.041 | 0.0002 $\pm$ 0.319 | 0.358 $\pm$ 0.747 | 1280 | ReLu |
| NasNetLarge | -2.080e-07 $\pm$ 0.026 | 4.424e-05 $\pm$ 0.040 | 0.198 $\pm$ 0.533 | 4032 | ReLu |
| NasNetMobile | -3.336e-07 $\pm$ 0.039 | 0.0001 $\pm$ 0.066 | 0.382 $\pm$ 4.389 | 1056 | ReLu |
| ResNet50 | 3.774e-07 $\pm$ 0.033 | -4.881e-08 $\pm$ 0.009 | 0.546 $\pm$ 0.871 | 2048 | ReLu |
| ResNet101V2 | 6.668e-06 $\pm$ 0.027 | 0.0016 $\pm$ 0.292 | 39.97 $\pm$ 167.8 | 2048 | ReLu |
| ResNet152V2 | 1.038e-05 $\pm$ 0.026 | 0.0016 $\pm$ 0.287 | 94.08 $\pm$ 187.4 | 2048 | ReLu |
| ResNet50V2 | 8.014e-07 $\pm$ 0.028 | 0.0011 $\pm$ 0.292 | 19.91 $\pm$ 74.65 | 2048 | ReLu |
| IncResNetV2 | -3.060e-05 $\pm$ 0.037 | -0.0012 $\pm$ 0.230 | 106.8 $\pm$ 124.9 | 1536 | ReLu |
| XCeption | -3.246e-06 $\pm$ 0.055 | 0.0008 $\pm$ 0.281 | 2.974 $\pm$ 13.41 | 2048 | ReLu |
| EffB0 | -7.495e-05 $\pm$ 0.068 | -5.143e-05 $\pm$ 0.058 | 0.065 $\pm$ 0.321 | 1280 | Swish |
| EffB1 | -5.647e-05 $\pm$ 0.063 | -4.343e-05 $\pm$ 0.045 | 0.056 $\pm$ 0.313 | 1280 | Swish |
| EffB2 | -7.152e-05 $\pm$ 0.059 | -4.153e-05 $\pm$ 0.054 | 0.019 $\pm$ 0.260 | 1408 | Swish |
| EffB3 | -6.323e-05 $\pm$ 0.054 | -3.547e-05 $\pm$ 0.046 | 0.010 $\pm$ 0.252 | 1536 | Swish |
| EffB4 | -3.106e-05 $\pm$ 0.050 | -3.138e-05 $\pm$ 0.057 | -0.039 $\pm$ 0.194 | 1792 | Swish |
| EffB5 | -2.043e-05 $\pm$ 0.049 | -2.738e-05 $\pm$ 0.055 | -0.036 $\pm$ 0.170 | 2048 | Swish |
| EffB6 | -8.656e-06 $\pm$ 0.046 | -2.691e-05 $\pm$ 0.071 | -0.043 $\pm$ 0.135 | 2304 | Swish |
| EffB7 | -9.562e-06 $\pm$ 0.041 | -2.441e-05 $\pm$ 0.060 | -0.041 $\pm$ 0.136 | 2560 | Swish |

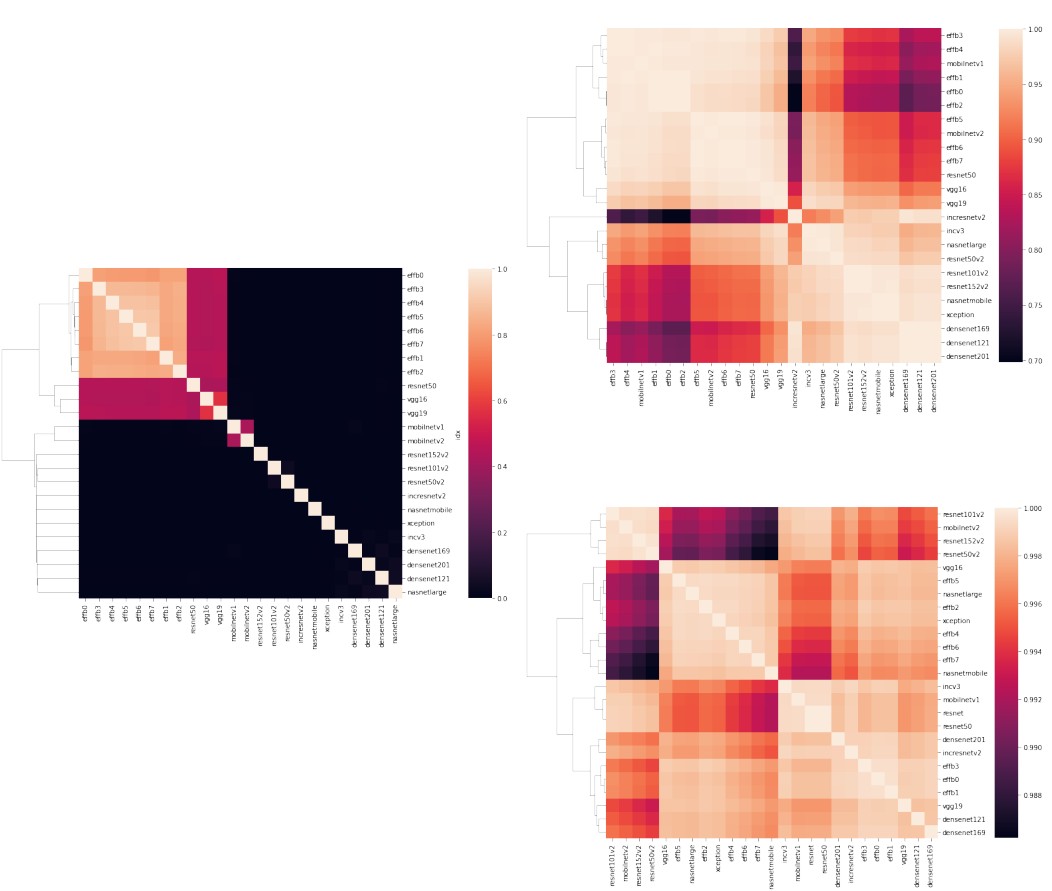

Figure 6: The similarity of twenty-four neural networks trained on the ImageNet data set. The base-line (left) is pair-wise fidality between the employed models compared to a similarity using Gromov-Wasserstein distance on the object (top right) and class (bottom right) view.

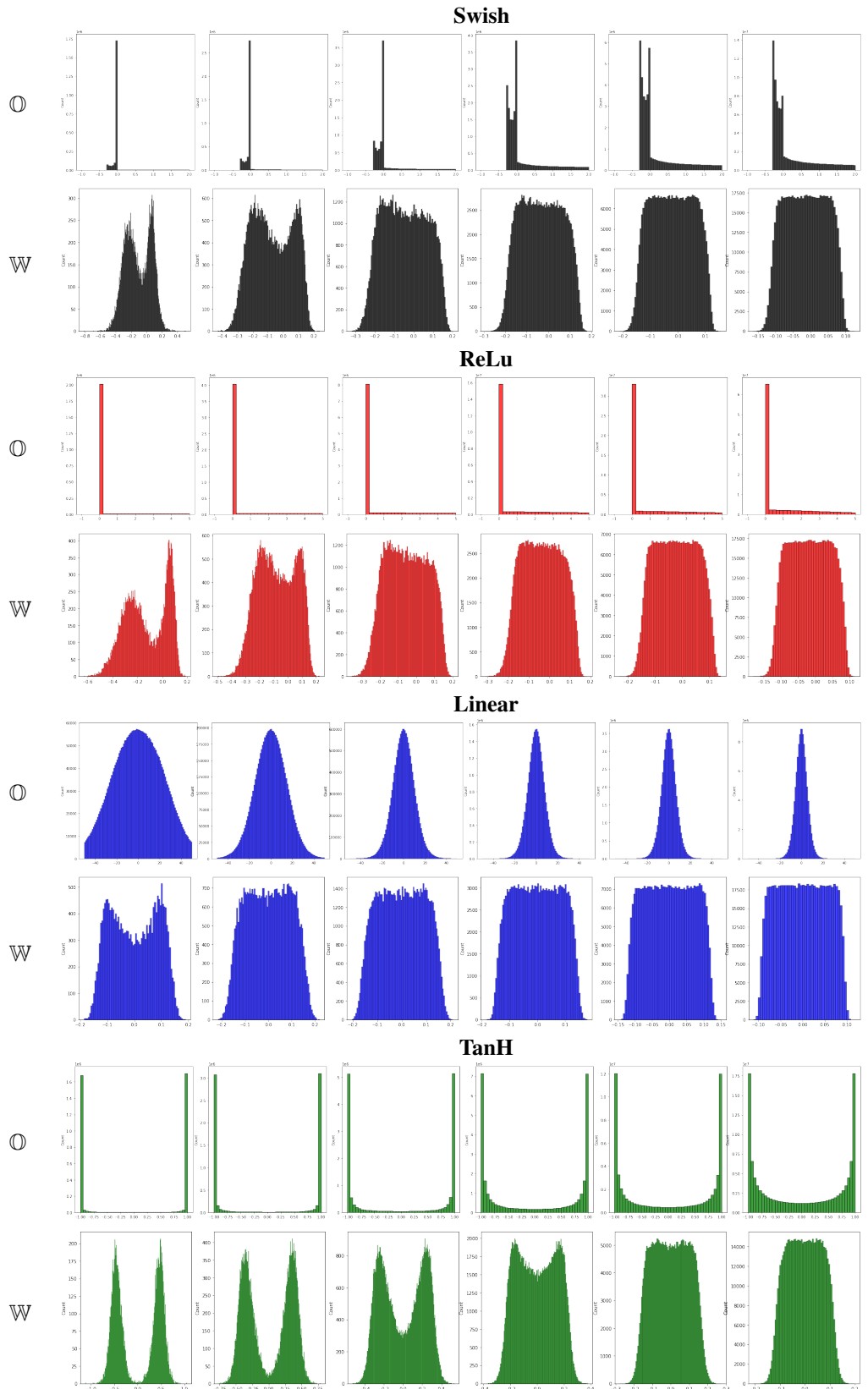

Figure 7: These are the value distributions for the object ($\mathbb{O}$) and class ($\mathbb{W}$) view for ten runs using the Fruits-360 data set and the swish, relu, linear and tanh activation functions. The last hidden layer of size $2^4$ (first column) to $2^9$ (last column).

Table 7: Results for the ablation study on the influence of the activation function and number of neurons on the quality of the (Symbolic) object/class view. The measure the quality in terms of fidelity of nearest neighbor classification to the original model, see V-Fid and SV-Fid for the symbolic case.

| | $2^4$ | $2^5$ | $2^6$ | $2^7$ | $2^8$ | $2^9$ |
|---|---|---|---|---|---|---|
| **Swish** | | | | | | |
| $\delta_\mathbb{O} = 0$ Split | 57.1/62.9 | 52.8/47.2 | 49.2/50.8 | 44.5/55.5 | 45.6/54.4 | 45.0/55.0 |
| $\delta_\mathbb{W} = 0$ Split | 65.4/44.6 | 65.0/35.0 | 62.0/38.0 | 60.1/39.9 | 57.2/43.8 | 56.5/43.5 |
| Model Acc | 93.5± 0.8 | 94.5± 0.5 | 95.3± 0.3 | 95.1± 0.3 | 95.4± 0.4 | 95.1± 0.5 |
| V-Fid | 99.5± 0.4 | 99.9± 0.0 | 99.9± 0.0 | 99.9± 0.0 | 99.9± 0.0 | 99.9± 0.0 |
| SV-Fid | 77.1± 9.2 | 88.8± 1.3 | 89.6± 1.6 | 88.8± 1.4 | 88.4± 0.9 | 86.7± 1.6 |
| **ReLu** | | | | | | |
| $\delta_\mathbb{O} = 0$ Split | 55.1/44.9 | 55.1/44.9 | 54.7/45.3 | 53.3/46.7 | 55.3/44.7 | 54.1/45.9 |
| $\delta_\mathbb{W} = 0$ Split | 66.3/33.7 | 66.7/33.2 | 64.0/36.0 | 61.6/38.4 | 58.9/41.1 | 58.2/41.8 |
| Model Acc | 93.7± 0.4 | 94.5± 0.5 | 94.9± 0.5 | 94.9± 0.5 | 95.0± 0.4 | 94.8± 0.6 |
| V-Fid | 99.7± 0.0 | 99.8± 0.0 | 99.9± 0.0 | 99.9± 0.0 | 99.9± 0.0 | 99.9± 0.0 |
| SV-Fid | 79.9± 3.7 | 89.0± 1.2 | 90.0± 1.2 | 89.5± 1.1 | 89.0± 1.5 | 88.2± 1.5 |
| | $2^4$ | $2^5$ | $2^6$ | $2^7$ | $2^8$ | $2^9$ |
| **Linear** | | | | | | |
| $\delta_\mathbb{O} = 0$ Split | 49.8/50.2 | 49.6/50.4 | 49.5/50.5 | 49.9/50.1 | 49.9/50.1 | 49.9/50.1 |
| $\delta_\mathbb{W} = 0$ Split | 49.7/50.3 | 49.3/50.7 | 49.7/50.3 | 50.0/50.0 | 50.0/50.0 | 50.0/50.0 |
| Model Acc | 85.3± 0.6 | 88.8± 0.7 | 89.9± 1.0 | 92.0± 0.5 | 91.8± 0.9 | 91.5± 0.9 |
| V-Fid | 99.9± 0.0 | 99.9± 0.0 | 99.9± 0.0 | 99.9± 0.0 | 99.9± 0.0 | 99.9± 0.0 |
| SV-Fid | 55.5± 1.9 | 64.7± 1.4 | 68.3± 2.6 | 74.8± 1.7 | 78.2± 2.1 | 81.5± 1.1 |
| **Tanh** | | | | | | |
| $\delta_\mathbb{O} = 0$ Split | 49.7/50.3 | 49.7/50.3 | 49.8/50.2 | 49.9/50.1 | 50.0/50/0 | 49.9/50.1 |
| $\delta_\mathbb{W} = 0$ Split | 49.9/50.1 | 49.8/50.2 | 49.8/50.2 | 50.0/50.0 | 49.9/50.1 | 50.0/50.0 |
| Model Acc | 90.5± 0.8 | 94.3± 0.5 | 94.7± 0.5 | 94.9± 0.4 | 95.0± 0.4 | 94.8± 0.3 |
| V-Fid | 98.3± 0.5 | 99.5± 0.1 | 99.7± 0.0 | 99.7± 0.0 | 99.8± 0.0 | 99.8± 0.0 |
| SV-Fid | 94.3± 1.4 | 97.4± 0.4 | 97.7± 0.4 | 97.6± 0.1 | 97.8± 0.2 | 97.6± 0.2 |

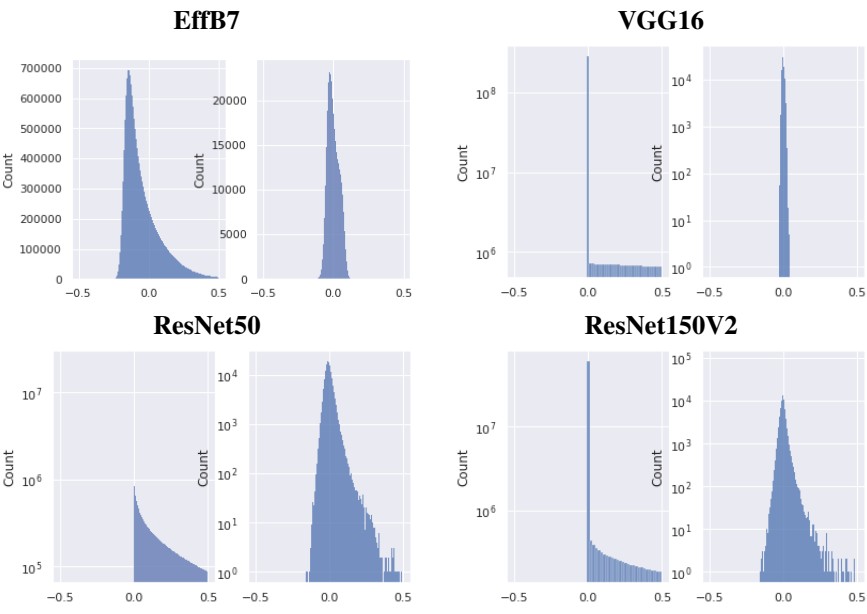

Figure 8: The value distributions for the object/class view for the EffB7, VGG16, ResNet50 and Resnet150V2 Model.

Table 8: The fidelity between twenty-four neural networks and their symbolic object/class view using nearest neighbor and cosine similarity for classification.

| | 1NN | Cos | Class Sep | Activation |
|---|---|---|---|---|
| VGG16 | 0.552 | 0.552 | 1.0 | ReLu |
| VGG19 | 0.5672 | 0.5672 | 1.0 | ReLu |
| IncV3 | 0.000 | 0.000 | 1.0 | ReLu |
| DenseNet121 | 0.000 | 0.000 | 1.0 | ReLu |
| DenseNet169 | 0.001 | 0.001 | 1.0 | ReLu |
| DenseNet201 | 0.000 | 0.000 | 1.0 | ReLu |
| MobilNetV1 | 0.036 | 0.036 | 1.0 | ReLu |
| MobilNetV2 | 0.305 | 0.305 | 1.0 | ReLu |
| NasNetMobile | 0.004 | 0.004 | 1.0 | ReLu |
| NasNetLarge | 0.009 | 0.009 | 1.0 | ReLu |
| ResNet50 | 0.007 | 0.000 | 1.0 | ReLu |
| ResNet50V2 | 0.001 | 0.003 | 1.0 | ReLu |
| | 1NN | Cos | Class Sep | Activation |
| ResNet101V2 | 0.012 | 0.012 | 1.0 | ReLu |
| ResNet152V2 | 0.000 | 0.000 | 1.0 | ReLu |
| IncResNetV2 | 0.000 | 0.000 | 1.0 | ReLu |
| XCeption | 0.220 | 0.220 | 1.0 | ReLu |
| EffB0 | 0.758 | 0.758 | 1.0 | Swish |
| EffB1 | 0.813 | 0.813 | 1.0 | Swish |
| EffB2 | 0.869 | 0.869 | 1.0 | Swish |
| EffB3 | 0.898 | 0.898 | 1.0 | Swish |
| EffB4 | 0.929 | 0.929 | 1.0 | Swish |
| EffB5 | 0.935 | 0.935 | 1.0 | Swish |
| EffB6 | 0.951 | 0.951 | 1.0 | Swish |
| EffB7 | 0.957 | 0.957 | 1.0 | Swish |

Table 9: The size of the concept lattices of the symbolic views in Table 4 (see $|N| = 16$) and for 32 neuron (see $|N| = 32$), for all (see All column) and only positive attributes (see Pos column).

| Model | $|N| = 16$ | | $|N| = 32$ | |
|---|---|---|---|---|
| | All | Pos | All | Pos |
| Base | 130969 | 6517 | 3192044 | 155416 |
| ResNet50 | 133130 | 5872 | 3803799 | 165009 |
| VGG16 | 126487 | 5200 | 3498829 | 193516 |
| IncV3 | 134100 | 5670 | 3782226 | 198152 |
| EffB0 | 132403 | 6573 | 3767964 | 150884 |

