# OpenReview forum: "Formal Conceptual Views in Neural Networks"
_ICLR.cc/2023/Conference — Submitted to ICLR 2023_

### Official Review · Reviewer_i9L7 · 2022-10-18

**Confidence:** 3
**Correctness:** 4
**Technical Novelty And Significance:** 3
**Empirical Novelty And Significance:** 3
**Recommendation:** 6

**Clarity, Quality, Novelty And Reproducibility:**

The description of the method is clear, though sometimes knowledge is unduly assumed, e.g. the definition of the Gromov-Wasserstein distance. The codebase looks clean and well-commented. The methods are put in context of related work and from this perspective appear to be novel, but as someone who is not so familiar with the literature, it is difficult for me to be completely confident on the novelty of the methods.

**Strength And Weaknesses:**

Strengths
- The paper tackles a challenging problem, that of neural network interpretability, which is of considerable interest to the ML community.
- The derivations of the concept views are clearly explained and easy to follow.
-  A key strength of the paper is that the conceptual views (in particular the symbolic view) are shown to enable several different types of analyses as summarised above: eg a similarity metric between different networks, decomposition into a formal concept lattice, and derivation of human-interpretable propositional statements when combined with background knowledge.
- A number of limitations of the methods are clearly outlined in the conclusion.

Questions / opportunities for improvement
- It is unclear to what extent the methods of the paper would apply outside of networks trained on classification tasks, e.g. on regression or even a stochastic policy network for reinforcement learning tasks. Could the framework easily be extended beyond classification tasks / would it even make sense?
- The symbolic view relies on a user-defined threshold that needs to be defined for the object view and class view - it’s not clear how the effectiveness/usefulness of the symbolic view depends on this choice, or what is a good way to make this choice in advance - it seems a bit arbitrary.
- In section 4.2, it would be nice to have more discussion of how we can interpret the clustering results - e.g. what could explain the difference between the clusters in the object view and the class view?
- In section 4.2, what is the rationale for shrinking the penultimate hidden layer as you grow the last hidden layer?
- For the similarity metric between networks, it would be good to have at least a high level description of what the Gromov-Wasserstein distance is for the reader who is not aware of it.
- As acknowledged by the authors, the method for deriving human-interpretable explanations from the conceptual views requires in addition the existence of domain-specific background knowledge, which limits the general applicability of the method.

Minor
- Section 4, Line 3  and in 4.1 “the in Section 3 introduced pseudo metric space” -> “the pseudo metric space introduced in Section 3”

**Summary Of The Paper:**

This paper proposes two methods for deriving ‘conceptual views’ of neural networks, a ‘many-valued’ view and a ’symbolic’ view, that allow for human-interpretable analyses into the knowledge contained in the network. Each view is constructed using the combination of a neural network (trained on a classification task) and a set of datapoints, and is a concatenation of an object-based view that corresponds to the activations of the last hidden layer for each datapoint, and a class view which corresponds to the weights from the last hidden layer to the output neurons. Via experiments using a variety of networks trained on Imagenet and Fruits-360 respectively, it is shown how one or both of the conceptual views can (i) be used to define a similarity metric between different neural networks, (ii) be used to represent a network with a formal concept lattice, which depicts a hierarchical clustering of features, and (iii) be used to derive human-interpretable propositional statements when combined with background knowledge of the dataset using subgroup discovery.

**Summary Of The Review:**

Overall, the paper does a good job of motivating and describing the conceptual views it proposes for neural network intepretability, and a range of applications of the views are described and demonstrated with experiments. The key weakness is that the general applicability of the methods is questionable, for instance it only seems to be defined for classification networks, and the human interpretability relies on the existence of domain-specific background knowledge.

---

### Official Review · Reviewer_r4Ld · 2022-10-24

**Confidence:** 4
**Correctness:** 2
**Technical Novelty And Significance:** 2
**Empirical Novelty And Significance:** 2
**Recommendation:** 3

**Clarity, Quality, Novelty And Reproducibility:**

For me this paper fails to be clear to a degree that I found it incomprehensible. Obviously there are papers that I struggle to understand and need to read multiple times, but there were so many strange statements that I was not encouraged to persist---for me, a demonstration of a lack of quality.  There was some novelty and it maybe that the work deserves publication, but not in this form.  I presume the work was reproducible.

**Strength And Weaknesses:**

The strength is that a new approach is proposed that is potentially interest.

I am afraid the weakness, for me at least, is that the paper is almost incomprehensible.  The paper introduces so many new concepts without really explaining what any of them mean or why they are relevant.  There are a huge number of statements that are incomprehensible to me.  Partly this is linguistic (some statements are difficult to parse), but much more there is no real effort to explain what the author means.  It comes across to me as a stream of consciousness rather than a well argued and justified thesis.  I am prepared to accept that this may be because the authors are much more knowledgable and clever that this reviewer, however, there is a necessity to explain ideas in a way that is comprehensible to a general readership.  I am afraid this paper fails this test in my opinion.  For this work to be publishable would, in my view, require a complete rewrite where all the ideas were fully explained.

**Summary Of The Paper:**

The paper covers a novel approach to understanding neural networks in terms of an "object view" (the activations of the network given the input representation of an object) and the "class view" (the set of weights between the neurons and the class outputs----I think).  In addition the paper introduces an intermediate representation using some known attributes called the conceptual representation.  The paper carries out a lot of empirical experiments using pretrained trained models on ImageNet and networks that are trained on Fruit-360.

**Summary Of The Review:**

For me, the paper was too full of poorly explained ideas.  It lacked a precision in writing but also an appreciation of the reader.  I accept that novel ideas are difficult to communicate, but for me the authors did not try to make their work understandable.  It is very difficult to make a judgement on what the authors did.  I have some scepticism about the utility of their approach, but ultimately I found it hard to care because the paper didn't seem to want to explain itself.

---

### Official Review · Reviewer_xUGn · 2022-10-26

**Confidence:** 3
**Correctness:** 3
**Technical Novelty And Significance:** 3
**Empirical Novelty And Significance:** 3
**Recommendation:** 3

**Clarity, Quality, Novelty And Reproducibility:**

Novelty and Quality: Due to the unclear motivation and application, it is hard to evaluate the main contribution. : It is hard to
Reproducibility: The author provides related code, which is good.
Clarity: This paper is a rushed version with many writing errors and missing explanations.

**Strength And Weaknesses:**

**Strength**

1 This paper shows two new views to help humans understand the learned knowledge of neural networks, which is interesting.

2 The paper conducted many experiments to support its claim.

**Weakness**

1 The motivation and application of the proposed method need to be clarified. Why do we need a Formal conceptual view to understand the knowledge of Neural Networks? The author uses some abstract words and sentences, e.g., "grasp deeper insights into the knowledge that is captured," which is hard to understand the motivation and application. In addition, it is hard to understand the goal of each step in the method section.

2) Why is only the last hidden layer important for the Many-valued Conceptual View? From a network dissection perspective, the last several layers may have a similar function: they are sensitive to high-level concept understanding. Why are the rest layers ignored? Also, how to extend the explanation to convolutional layers (kernels), transformer-based layers or neural networks trained with self-supervised learning (contrastive loss)? The author only explains the vanilla MLP model.

3 Use a large number of notations while some of them lack explanation. For instance, what is the meaning of it? What is the meaning of | |?

4 Many typos and grammar errors which obstacles the understanding. For instance, Definition1: C its..., N = {} the..., Section 3.1 "As a final remark before we introduce the symbolic conceptual view on NN we want to point out a simple but powerful observation." no comma at all. Section 4. "First, we evaluate the suitability of the in Section 3 introduced pseudo metric space by a classification task. "

**Summary Of The Paper:**

This paper provides two views to globally explain the learned knowledge of NN to humans. Specifically, the "many-valued" view explains each decision as a combination of object representation and classification, which focuses on the last hidden layer. The symbolic view can be used to represent NN models with formal concept lattices. This paper uses two main experiments to support the claim.

**Summary Of The Review:**

See above

---

### Official Review · Reviewer_PGXP · 2022-10-27

**Confidence:** 3
**Correctness:** 3
**Technical Novelty And Significance:** 1
**Empirical Novelty And Significance:** 1
**Recommendation:** 3

**Clarity, Quality, Novelty And Reproducibility:**

The novelty / technical contribution of the paper is very limited as far as i can tell. While the contribution is explained in terms of symbolic schema, what it does is in other words is that it discretizes the representations of different datapoints, and considers similarity in terms of the weights operating in the last layer for classification and the features computed on the inputs.

The paper is very hard to understand and is full of notation without clarity of what the underlying ideas are. Especially Sec. 5 which is potentially the most interesting part of the paper is very difficult to understand. It is not clear at all why S_m the interpretable features can be used along with the symbolic neural view of the network.


**Writing**
- It might make more sense to explain how the conceptual views can be used to compute similarity before discussing how the approach is different from prior works
- Writing is very fragmented, it is not clear at all how the thresholding operation allows one to utilize human interpretable features for representation. More details are said to be given in Sec. 5 but the details are actually only given Sec. 3.1 while Sec. 5 discusses an application.


**Strength And Weaknesses:**

+ The paper reports nice ablations and experimental results comparing different design choices
+ The paper tackles an important problem of neural network interpretability, which is as relevant as ever and of great practical interest
+ The paper is looking at a number of tasks ranging from comparing two neural networks for similarity, interpreting what a trained network has learned, and quality of function approximation with the discrete representation


**Summary Of The Paper:**

The paper looks to interpret neural networks by examining the representations and the weights. The paper does a discretization of both of these and indexes them via a look up to create what it calls the symbolic conceptual view of a neural network. This conceptual view is then evaluated on various tasks. Experiments show that the conceptual view can approximate a neural network well, be used to compute similarity between networks and do “abductive learning” which as far as I can tell is about interpreting a trained neural network by giving human understandable forms to the symbolic conceptual view.


**Summary Of The Review:**

The paper is very hard to read, and understand. From what I can understand the technical novelty is very limited and uninteresting.

---

### Author Response · Authors · 2022-11-16
**Authors' comments to the reviews**

We agree with the reviewers that our notion to employ relational structures may be difficult to understand at first hand. This is particularly true if one is not familiar with ordinal data science. The proposed discretization is only one of several steps we have introduced to derive relational and later ordinal conceptual structures. We agree that this step seems natural and simple. This is by design to ensure that the derived conceptual structures are understandable to, potentially untrained, human readers. In our experiments, we have empirically demonstrated that this approach is consistent and meaningful with respect to classification tasks.
Although there is much literature on the interpretation of ordinal, and in particular conceptual, structures, we agree that our work would benefit from more explanations. This is also true for the various number of common machine learning methods we included, e.g., the Gromov-Wasserstein distance. We plan to include these in a longer version.

---

### Decision · Program_Chairs · 2023-01-20

**Decision:**

Reject

**Justification For Why Not Higher Score:**

Uniformly low scores. Authors did not really attempt a rebuttal.

**Justification For Why Not Lower Score:**

N/A

**Metareview: Summary, Strengths And Weaknesses:**

This paper is concerned with given a human analyst the ability to understand the knowledge captured in a neural network. They pursue this by thresholding weights and representations, producing a discrete representation of networks and data. They then propose to study networks by studying these discrete objects.

The reviewers found the paper thoroughly confusing and could not discern what the contributions were or what the impact would be. I would agree that it is both hard to read and very hard to discern what is achieved. No reviewer argued for acceptance. I think the paper suffers from having ill-defined goals. The approach advocated here simply shifts the problem to another space (yet with less fidelity).

**Summary Of Ac-Reviewer Meeting:**

N/A